# Nanotopographical 3D-Printed Poly(ε-caprolactone) Scaffolds Enhance Proliferation and Osteogenic Differentiation of Urine-Derived Stem Cells for Bone Regeneration

**DOI:** 10.3390/pharmaceutics14071437

**Published:** 2022-07-08

**Authors:** Fei Xing, Hua-Mo Yin, Man Zhe, Ji-Chang Xie, Xin Duan, Jia-Zhuang Xu, Zhou Xiang, Zhong-Ming Li

**Affiliations:** 1Orthopedic Research Institute, Department of Orthopedics, West China Hospital, Sichuan University, No. 37 Guoxue Lane, Chengdu 610041, China; xingfei@scu.edu.cn (F.X.); xiangzhou@scu.edu.cn (Z.X.); 2College of Polymer Science and Engineering, State Key Laboratory of Polymer Materials Engineering, Sichuan University, Chengdu 610065, China; huamoyinscu@163.com (H.-M.Y.); zmli@scu.edu.cn (Z.-M.L.); 3Animal Experiment Center, West China Hospital, Sichuan University, Chengdu 610041, China; zheman@wchscu.cn; 4Laboratoire Roberval, FRE UTC-CNRS 2012, Sorbonne Universités, Université de Technologie de Compiègne, Centre de Recherche Royallieu, CS60319, CEDEX, 60203 Compiègne, France; xjctjpu@163.com

**Keywords:** nanotopographical, bone regeneration, stem cell, 3D printing, bone repair

## Abstract

3D-printing technology can be used to construct personalized bone substitutes with customized shapes, but it cannot regulate the topological morphology of the scaffold surface, which plays a vital role in regulating the biological behaviors of stem cells. In addition, stem cells are able to sense the topographical and mechanical cues of surface of scaffolds by mechanosensing and mechanotransduction. In our study, we fabricated a 3D-printed poly(ε-caprolactone) (PCL) scaffold with a nanotopographical surface and loaded it with urine-derived stem cells (USCs) for application of bone regeneration. The topological 3D-printed PCL scaffolds (TPS) fabricated by surface epiphytic crystallization, possessed uniformly patterned nanoridges, of which the element composition and functional groups of nanoridges were the same as PCL. Compared with bare 3D-printed PCL scaffolds (BPS), TPS have a higher ability for protein adsorption and mineralization in vitro. The proliferation, cell length, and osteogenic gene expression of USCs on the surface of TPS were significantly higher than that of BPS. In addition, the TPS loaded with USCs exhibited a good ability for bone regeneration in cranial bone defects. Our study demonstrated that nanotopographical 3D-printed scaffolds loaded with USCs are a safe and effective therapeutic strategy for bone regeneration.

## 1. Introduction

Treatment of large bone defects resulting from trauma, cancer, infection, congenital malformation, or surgical resection is a challenge for clinical doctors [1]. Autografting bone transplantation is the gold standard for the treatment of large bone defects [2]. However, many complications arise with this treatment approach, such as graft resorption, limited sources, donor-site morbidity, and infection [3]. Currently, the alternative bone augmentation materials used in clinical work can be grouped into three categories: cellular bone matrices, growth factor-enhanced bone grafts, and peptide-enhanced xeno-hybrid bone grafts [4,5]. All these bone substitutes possess a good capacity for bone regeneration. However, the long-term safety and efficiency of these bone substitutes still require further research. On the other hand, with the rapid development of technologies in regenerative medicine, bone grafts constructed by tissue-engineered methods are also rapidly becoming promising alternatives for bone regeneration [6]. Tissue regeneration is performed by implanting cells and biomaterials into the body, which rebuilds tissues and supports its native self-healing abilities to promote tissue growth [7]. In addition, tissue engineering has made great strides over the past few decades in advancing scaffold design and architecture with the fabrication of many constructs that mimic the biological, mechanical, and chemical characteristics of the host tissue [8]. The scaffolds constructed by bone tissue engineering can induce osteoconduction and integration, provide mechanical stability, and either integrate into the bone structure or degrade and then be excreted by the body [9]. Currently, various tissue-engineered bone substitutes have been fabricated for the treatment of bone defects [2].

Recently, many studies have confirmed the existence of stem cells in urine and named urine-derived stem cells (USCs) [10,11]. USCs share similar biological properties with MSCs and possess the capacities of robust proliferation and multi-potential differentiation [12]. In addition, due to their simple, cost-effective, and non-invasive separation procedure, USCs might be a promising alternative kind of seed cell for stem cell-based therapy. In our previous studies, we have demonstrated that USCs could be used as seed cells for bone and cartilage regeneration [13,14,15,16,17]. In addition, our previous study also found that USCs had a better capacity for proliferation, colony-forming, and migration compared to bone marrow mesenchymal stem cells (BMSCs) in the same passage [18]. However, compared to BMSCs, the USCs in the same passage have lower osteogenic, adipogenic, and chondrogenic abilities [18]. Recently, many researchers have focused on improving the osteogenic capacity of USCs to make them more suitable for bone tissue regeneration.

In the process of tissue repair, scaffolds, as carriers of various cells, play an important role in regulating the biological behavior of cells. The surface characteristics of scaffolds, mainly include physical and biochemical cues, can manipulate the fate of stem cells, such as proliferation, differentiation, migration, and adhesion. At the same time, the cells are able to sense the topographical and mechanical cues of surface of scaffolds by mechanosensing and mechanotransduction [19,20]. In addition to specific biochemical cues, these physical cues, especially topographical cues, are more general regulators of nearly all cellular behavior and functions and the efficiency of biochemical stimulation [21,22]. During the process of cell adhesion, the cells control myosin motors to generate intracellular force, which further regulate the biological behaviors of cells at the adhesive interface by affecting the nuclear mechanics and the transcription factor activity. This mechanical process causes aligned structural changes in cell morphology, which in turn leads to cell polarization. The polarization plays an essential role in constructing the anisotropic organization of various tissues and has been proved to enhance cell proliferation and osteogenic differentiation [23,24]. Therefore, in order to induce cell polarization, many researchers fabricated various tissue-engineered scaffolds with aligned topologies, which include wrinkle, pillar, channel, groove, pits, and fiber [25,26,27]. In the field of bone regeneration, fabricating the scaffolds with aligned nanotopologies is beneficial for modulating and loading seed cells.

Compared with traditional tissue engineering scaffold manufacturing methods, 3D printing technology has the advantages of high designability and high repeatability [28,29]. When tissue engineering settings are considered, 3D printing allows using several biomaterials such as biopolymers, to manufacture tissue-like 3D micro- and macro-structures containing biochemicals and even living cells [7,30]. However, the current 3D printing technology cannot control the nanotopological morphology of 3D-printed scaffolds. Simultaneously, the surface topological morphology of most 3D-printed scaffolds fabricated by polymer materials exhibits flat topography, resulting in low biological activity [31,32]. Constructing suitable topological morphology on the surface of 3D-printed scaffolds might enhance its biological activity. In this study, we aimed to fabricate a 3D-printed poly(ε-caprolactone) (PCL) scaffold with specific nanotopological morphology and loaded with USCs for application in bone regeneration.

## 2. Materials and Methods

### 2.1. Fabrication of TPS

The 3D-printed PCL scaffolds were fabricated by 3D printer (CR3040, Shenzhen Chuangxiang Industrial Co., Ltd., Shenzhen, China), which is based on melt extrusion deposition (MED™) printing method. PCL (Esun 600C) with a viscosity-averaged molecular weight of 6 × 10^4^ g/mol was purchased from Shenzhen Guanghua Weiye Industrial Co., Ltd., Shenzhen, China. PCL pellets were vacuum dried at 50 °C for 12 h, and then were put into single-screw extruder to form PCL filaments with a diameter of 1.75 mm. The PCL filaments were melted into the extruder nozzle, which then deposited the melted material onto the platform. The 3D-printed PCL scaffold was designed into a cylindrical grid. The printing conditions were set as follows: diameter of the printing needle = 0.5 mm, movement speed of the needle = 0.5 mm/s, printing temperature = 70 °C, pause time between printing two layers = 0.1 s. The distance between neighboring fiber centers was 1 mm, and the fibers were arrayed orthogonally.

Acetic acid (Chengdu Kelong Chemical Reagent Factory, Chengdu, China) and distilled water were mixed in a volume ratio of 77%/23%. A certain amount of PCL particles was fully dissolved into the prepared mixed solution at 60 °C and cooled to ambient temperature. Then, the 3D-printed PCL scaffolds were immersed in the as-prepared mixed solution for 5 min. Finally, 3D-printed PCL scaffolds were air-dried at 25 °C to form nanotopological structure via epitaxial crystallization, which is named TPS. The bare 3D-printed PCL scaffold (BPS) was prepared as a control.

### 2.2. Characterization of Substrate Surface of TPS and BPS

Field-emission scanning electronic microscopy (SEM, Nova NanoSEM450, FEI, Hillsboro, OR, USA) and atomic force microscopy (AFM, Afm+_nanoTA, Anasys Instruments Inc., Goleta, CA, USA) were used to observe the morphology. Surface chemistry was studied by Fourier transform infrared (FTIR) spectroscopy (Nicolet 6700, Thermal Scientific, Alvarado, TX, USA) in a wavenumber range of 650–2000 cm^−1^. X-ray photoelectron spectroscopy (XPS, XSAM800, Shimadzu-Kratos Ltd., Kanagawa, Japan) was performed to investigate the surface elements. Hemolysis assay was applied to evaluate the hemocompatibility of TPS and BPS. Additionally, the protein adsorption capacity of TPS and BPS was also tested by using a protein estimation kit (Beyotime, BCA Protein Assay Kit, Shanghai, China).

### 2.3. Isolation and Identification of USCs

USCs were isolated and cultured using the methods that were described previously [13,16]. Briefly, 200–250 mL urine samples were collected from healthy adult male donors and divided into 50 mL centrifuge tubes. Each 50 mL tube was centrifuged for 15 min at 1700 rpm and the supernatant was discarded. The pellet was resuspended and washed twice with PBS. Then, the cell pellets were resuspended in keratinocyte serum-free medium (KSFM) (Thermo Fisher Scientific Inc., Fremont, CA, USA) and embryo fibroblast medium (EFM) (Thermo Fisher Scientific Inc., Fremont, CA, USA) (1:1 ratio) and seeded into 24-well culture plates. The medium was changed every three days, and cells were passaged by using trypsin after reaching subconfluency. When at passage 4, USCs were harvested and washed twice with PBS, then stained with the following specific antihuman antibodies: CD29, CD90, CD45, HLA-DR (BD Biosciences, San Jose, CA, USA) [33,34,35]. When at Passage 4, USCs were incubated in human MSC osteogenic differentiation medium, which was Dulbecco’s modified eagle medium supplemented with 10% fetal bovine serum, 10 mM of β glycerol phosphate disodium, 0.2 μM dexamethasone, and 50 μg/mL ascorbic acid. Alizarin red S working solution and alkaline phosphatase staining were used to evaluate osteogenic differentiation of USCs. Additionally, growth curve of USCs was also generated by Cell Counting Kit-8 (CCK8, APExBIO Technology LLC, Houston, TX, USA).

### 2.4. Biological Behaviors of USCs on the Surface of TPS and BPS

We adjusted the cell density of USCs to 3 × 10^5^ mL and inoculated 100 μL of cell suspension on the surface of TPS and BPS. Live/Dead staining (Sigma, Milwaukee, WI, USA) and CCK-8 were performed to observe the cell proliferation. Additionally, we adjusted the cell density of USCs to 2 × 10^5^ mL and inoculated 100 μL of cell suspension on the surface of TPS and BPS. Phalloidin staining (Invitrogen, Carlsbad, CA, USA) was used to investigate the cytoskeleton. After culturing for 3 and 7 days, osteogenesis-related gene expression of USCs was investigated. Briefly, trypsin was utilized to collect USCs from BPS and TPS. Gene expression levels of runt-related transcription factor 2 (*RUNX2*), collagen type 1 alpha 1 (*COL1A1*), alkaline phosphatase (*ALP*), bone morphogenetic protein 2 (*BMP2*), osteocalcin (*OCN*), and osteopontin (*OPN*) were detected by real-time quantitative polymerase chain reaction (RT-qPCR). Glyceraldehyde-3-phosphate dehydrogenase (*GAPDH*) was used as a reference gene. The primer sequences of these genes are listed in Table 1. TRIzol solution (Invitrogen, Carlsbad, CA, USA) was utilized to extract total RNA from USCs, and QuantiTect Reverse Transcription Kit (Thermo Fisher Scientific Inc., Fremont, CA, USA) was used to reverse-transcribed RNA into cDNA. An SYBR Green Mix Kit (Roche Company, Berlin, Germany) was used to amplify the specific transcripts on the real-time fluorescence quantitative instrument. The PCR procedure was heat at 94 °C for 5 min followed by 40 cycles of 94 °C for 15 s, 55 °C for 30 s, and 72 °C for 30 s. Marker gene expression data were analyzed by the 2^−DDCq^ method using *GAPDH*. Additionally, the ability of apatite to form on the surface of BPS and TPS was examined by immersing in simulated body fluid (SBF) with ion concentrations nearly equal to those of human blood plasma [36].

### 2.5. In Vivo Bone Regeneration Evaluation of TPS Loaded with USCs

The animal study was approved by institutional ethical review board of our institution. Twenty adult healthy New Zealand White rabbits (2.5–3.0 kg in weight) were selected for establishing a model of cranial bone defect. After general anesthesia, a longitudinal incision along the midline of about 3 cm in length was performed by the surgeon, followed by incision of the skin and subcutaneous tissues in layers. After removal of the periosteum, two circular cranial bone defects with a diameter of 8 mm were created with a dental implant trephine. After implanting different groups of scaffolds, the wounds were carefully flushed with saline and closed layer by layer. The images of rabbit cranial bone defect preparation are shown in Figure 1. Twenty animals were randomly divided into 5 groups (TPS loaded with USCs, BPS loaded with USCs, TPS, BPS, blank control). In groups of TPS loaded with USCs and BPS loaded with USCs, the cell density of USCs was adjusted to 10^7^ mL and then the 100 μL of cell suspension was seeded onto the TPS and BPS, respectively. After cultured in osteogenic medium for 7 days, TPS loaded with USCs and BPS loaded with USCs were implanted into cranial bone defects. In groups of TPS and BPS, TPS and BPS were directly implanted into cranial bone defects. As a control, no scaffolds were implanted into cranial bone defects in group of blank control. All animals were sacrificed by an overdose of anesthesia at six and twelve weeks after surgery. Cranial bone defects from each group were assessed using Micro-CT, then cranial bone defects were fixed with neutral formalin solution for two weeks, and then removed calcium using the EDTA solution after cleaning by distilled water. Routine gradient dehydration, xylene transparency, paraffin embedding, section, hematoxylin and eosin (HE) staining (YEASEN, Shanghai, China), and Masson staining (YEASEN, China) were performed.

### 2.6. Statistical Analysis

All the values were reported as means ± standard deviation (SD). The statistical analysis of the differences between the study groups was used one-way analysis of variance (ANOVA). *p* values of <0.05 was considered statistically significant. All the statistical analyses were performed by SPSS Statistics version 22.0.

## 3. Results

### 3.1. Fabrication and Characterization of TPS

The two-dimensional schematic diagram of 3D-printed PCL scaffolds is shown in Figure 2A. The diameter of the inner fiber was 0.5 mm, and the distance between the inner fiber centers was 1 mm. The scaffold was designed into a four-layer grid-like object. Additionally, the fiber angle between adjacent layers in the scaffold is 90°. Figure 2B shows the procedure of surface topological modulation of 3D-printed PCL scaffolds. The general view of TPS and BPS exhibited a white grid cylindrical shape (Figure 2C). The SEM results revealed that the surface topological morphology of BPS was flat and smooth, while the surface topological morphology of TPS fabricated by surface epiphytic crystals presented with a layer of uniformly patterned nanoridges (Figure 2D). The AFM results also demonstrated that the surface topological morphology of BPS was flat, and the surface fluctuations are low. The surface fluctuation of TPS was high and the surface topological morphology was rough (Figure 2E). Additionally, the distribution of the nanoridge size and the periodic distance of TPS are shown in Figure 2F,G, respectively. The average nanoridge size of TPS was 67.46 ± 9.14 nm. The average periodic distance of TPS was 450.74 ± 50.20 nm. The FTIR results of BPS and TPS are shown in Figure 2H. The FTIR results of BPS before and after surface topological modulation demonstrated that the decorated and pristine substrates share the same characteristic IR bands with PCL, 1170 cm^−1^ (symmetric COC stretching), 1239 cm^−1^ (asymmetric COC stretching), 1294 cm^−1^ (C−O and C−C stretching), 1722 cm^−1^ (carbonyl stretching), 2865 cm^−1^ (symmetric CH_2_ stretching), and 2945 cm^−1^ (asymmetric CH_2_ stretching) [37]. XPS results showed that decorated and pristine substrates of 3D-printed scaffolds formed two characteristic peaks of carbon and oxygen elements at 285.33 and 523.34 eV, respectively, which demonstrated that the decorated nanoridges constructed by epitaxial crystallization had the same chemical composition as the substrate (Figure 2I). In terms of porosity and mean pore size of scaffolds, no significant differences were found between TPS and BPS (Appendix A). Additionally, the hemolysis assay was used to determine the hemolytic effect of TPS and BPS in our study, which is a requirement to be tested for blood-contacting bone substitutes [38]. When the hemolysis rate of the sample is >5%, it indicates the sample is hemolytic [38]. The hemolysis rates of TPS and BPS were 3.29 ± 1.00% and 2.32 ± 1.09% respectively, which demonstrated the good hemocompatibility of TPS and BPS (Figure 2J). No significant differences were observed between the normal medium and medium infiltrated by the TPS or BPS (Appendix A). The animal skin stimulation of BPS or TPS showed negative results (Appendix A). The adsorption of protein can play a valuable role in the subsequent cell behavior based on the fact that various proteins can participate in the different biological processes of cells. In our study, we found that the albumin adsorption ability of TPS was significantly higher than that of BPS (Figure 2J).

### 3.2. Isolation and Identification of USCs

USCs isolated from the urine of healthy adults present a morphology with rice grains (Figure 3A). After osteogenic differentiation of USCs, calcium nodules were observed by Alizarin Red staining (Figure 3B), and the alkaline phosphatase staining was also positive (Figure 3C). Figure 3D shows the results of flow cytometric analysis. Flow cytometric analysis demonstrated that the USCs were positive for MSC markers CD29 and CD90 [33], and negative for hematopoietic lineage marker CD45 [34], and immunocyte marker HLA-DR [35]. Additionally, the growth curve of USCs was typically S-shaped, including a slow growth period, a logarithmic growth period, and a plateau period (Figure 3E).

### 3.3. Biological Behaviors of USCs on the Surface of TPS and BPS

Figure 4A shows the results of Live/Dead staining. Live cells are stained green, and dead cells are stained red. The results found that most of USCs could survive on the surface of BPS and TPS. With the extension of the culture time, the number of living cells on the surface of the scaffold gradually increased. Additionally, the number of living cells on the surface of TPS was higher than that of BPS. The survival rates of USCs based on the results of Live/Dead staining are shown in Appendix A. The survival rate of TPS was significantly higher than that of BPS on day 5. CCK8 is used to quantitatively analyze cell proliferation on the surface of the scaffolds (Figure 4B). The results showed that the OD values of the TPS group were significantly higher than that of the BPS group from day 3 to day 9, which indicated the faster cell proliferation rate of the TPS group. The cytoskeleton of USCs on the surface of scaffolds was observed by phalloidin staining (Figure 4C). The F-actin inside the USCs is stained green, the distribution of which can show the cell shape. The USCs on the surface of TPS showed a long rod shape, while the USCs on the surface of BPS showed a round or oval shape. The cell lengths were calculated by the phalloidin staining results. The cell lengths of USCs on the surface of TPS were significantly longer than that of BPS (Figure 4D). The SEM image of scaffolds loaded with USCs is shown in Figure 4E. Compared with BPS, more extracellular matrix of USCs was deposited on the surface of TPS. The mineralization ability of the in vitro scaffolds was achieved by immersing the scaffolds in the SBF solution. Compared with BPS, a large amount of apatite formed on the surface of TPS after a period of immersion in SBF (Figure 4F). The osteogenic-related gene expression of USCs on the surface of the scaffold at different times is shown in Figure 4G. *RUNX2* is a member of the *RUNX* family of transcription factors and encodes a nuclear protein with a Runt DNA-binding domain. RUNX2 protein is essential for osteoblastic differentiation and skeletal morphogenesis and acts as a scaffold for nucleic acids and regulatory factors involved in skeletal gene expression [39]. The *RUNX2* expression of the TPS group was significantly higher than that of the BPS group from day 3 to day 7. COL1A1, an early marker of osteogenic differentiation, is a major extracellular matrix component of periodontal tissues that promotes osteoblast differentiation and mineralization [40]. The *COL1A1* expression of the TPS group was significantly higher than that of the BPS group from day 3 to day 7. BMP2 is a bone-inducing differentiation factor that causes mesenchymal stem cells (MSCs) to differentiate into osteoblasts [41]. ALP, produced by osteogenic cells, is one of the most reliable early markers for osteogenic differentiation [42]. OCN is considered a late marker of osteogenic differentiation and its expression at high levels indicates maturation and terminal differentiation of osteoblasts [43]. As another osteogenic marker, *OPN* expression is strictly controlled but pivotal for the expression of secondary late markers [44]. While no significant differences were found between the groups of TPS and BPS.

### 3.4. Radiologic Evaluation of Scaffolds after Transplantation In Vivo

Micro-CT was used to investigate the new bone formation in bone defects (Figure 5A). At six weeks after implantation, there was a very small amount of new bone formation at the edge of the bone defect in the blank group, while there was no new bone formation in the central area of the bone defect. In the bone defect of each scaffold group, there were strip-shaped new bone tissues of various lengths and thicknesses formed and distributed in the defect area. Additionally, the strip-shaped new bones in the bone defects of the groups of TPS and TPS loaded with USCs are more evenly distributed than the groups of BPS and BPS loaded with USCs. At twelve weeks after implantation, there was more new bone tissue at the edge of the bone defect compared with before in the blank control group, and the shape of the defect edge became irregular, but there was still no new bone formation in the central area of the bone defect. Compared with the BPS group, the volume of strip-shaped bone tissue in the TPS group was larger and the distribution of new bone tissue in the bone defect was more even. Additionally, the area of new bone in the TPS loaded with the USCs group was significantly higher than that of the other groups, and the thickness of the strip-shaped new bone was also significantly higher than that of the other scaffold groups.

According to the results of Micro-CT, BV/TV was calculated to quantitatively analyze the new bone volume of bone defects in animal models (Figure 5B). At six weeks and twelve weeks after implantation, the BV/TV of each scaffold group was significantly higher than that of the blank control group. The BV/TV of the TPS group was significantly higher than that of the BPS group at six and twelve weeks after surgery. Simultaneously, compared with the group of BPS loaded with USCs, the group of TPS loaded with USCs have a higher BV/TV after implantation. While no significant differences were found between groups of TPS, and BPS loaded with USCs.

### 3.5. Histological Evaluation of Scaffolds after Transplantation In Vivo

The HE staining results of the central area of the bone defects at six and twelve weeks after implantation are shown in Figure 6. All scaffolds were tightly integrated with the edge of the bone defect. The shape of the scaffolds in all groups had changed to varying degrees. A large amount of fibrous tissue is distributed inside the BPS at six weeks postoperatively. A very small amount of new bone tissue was formed in the marginal area of the scaffolds at twelve weeks postoperatively. A small amount of new bone tissue formed near the central area inside the TPS at six weeks after implantation. At twelve weeks after surgery, more bone tissue formed in the central area of TPS. In the group of BPS loaded with USCs, the new bone tissue is mainly distributed in the marginal area inside the scaffold. In the group of TPS loaded with USCs, new bone tissue was observed inside the scaffold at six weeks after the operation. Simultaneously, there was new bone tissue growing from the defect edge into the scaffolds. At twelve weeks after implantation, more new bone tissue is evenly distributed inside the scaffold in the group of TPS loaded with USCs. The HE staining results of liver and kidney tissues at six and twelve weeks after surgery showed that the histological morphology of liver lobules, glomeruli, and renal tubules in the four scaffold groups were the same as that of normal liver and kidney tissues (Figure 7).

The Masson staining results of all scaffold groups at six and twelve weeks after transplantation were shown in Figure 8A. The mineralized bone tissue was stained blue, while the osteoid, unmineralized bone tissue, was stained red [45]. The Masson staining results of new bone formation were consistent with that from HE staining. The bone tissue formed in the group of BPS loaded with USCs was mostly osteoid, while the bone tissue formed in the group of TPS loaded with USCs consisted of mineralized bone and osteoid. Additionally, according to the Masson staining results, new bone areas in bone defects were calculated for quantitative analysis (Figure 8B). The quantitative analysis results at six and twelve weeks after surgery showed that the new bone area in the group of TPS loaded with USCs was significantly higher than in the other three scaffold groups. The new bone area of the TPS group was significantly higher than that of the BPS group. In addition, no significant differences were found between groups of TPS, and BPS loaded with USCs.

## 4. Discussion

Urine, as a biological waste produced by the human body, contains a small population of USCs with self-renewal capacity and multidirectional differentiation [11]. Consistent with previous research, our study also demonstrated that USCs have a similar phenotype to mesenchymal stem cells (MSCs). HLA-DR, as one of class II HLA glycoproteins, often exists on the surface of antigen-presenting cells and takes part in triggering an immune reaction in vivo [46]. Our study demonstrated that USCs do not express HLA-DR, which might be the cause of the immunosuppressive effect of USCs in cell transplantation. In recent years, USCs have been generally used in tissue regeneration due to safe, low-cost, simple, and noninvasive isolation procedures [47]. Although the osteogenic differentiation of USCs in vitro has been demonstrated by many studies, the studies focusing on the bone regeneration of USCs in vivo are limited. Additionally, how to enhance the osteogenic ability of USCs in bone substitutes still needs further studies. Therefore, our study loaded USCs on the surface of scaffolds for bone defect repair. The results in vivo demonstrated that the scaffolds loaded with USCs have a better ability to repair bone defects in vivo.

As a new kind of scaffold manufacturing technology, 3D-printing technology has good application prospects in the construction of personalized scaffolds for the treatment of bone defects [48]. In recent years, many kinds of thermoplastic polymer materials, such as PCL and PLA, can be constructed into single or composite bone repair scaffolds for bone regeneration through fused deposition 3D-printing technology [49]. However, the current 3D-printing technology still has shortcomings in the fabrication of the surface topography of the 3D-printed scaffolds. The surface topography of most 3D-printed polymer scaffolds is flat and smooth [50,51]. After the scaffolds are implanted in the bone defects, biomimetic surfaces could resemble native extracellular matrixes and provide an optimum microenvironment for controlling cell behaviors [52]. As one of the surface chemical and physical cues, the surface topography of scaffolds could effectively regulate the expansion and differentiation of human stem cells in vitro and in vivo by motivating biophysical signals [53,54]. The previous study demonstrated that nanopillar structures with heights of 15 nm resulted in promoting bone regeneration on the surface of titanium [55]. Additionally, formation of nanotubes with heights of 15 nm on the substrates promotes cell adhesion, proliferation, mineralization levels, and osteocalcin expression [56]. Dalby et al. demonstrated that nanopits with a controlled structure could effectively enhance osteogenic differentiation of human mesenchymal stem cells [57]. Therefore, micro/nano patterning processing on the surface of the 3D-printed scaffold may help improve the 3D-printed scaffold’s ability to regulate the fate of seed cells. In our previous study, we successfully fabricated a layer of lamellar nanosheets on the surface of two-dimensional PCL substrates, which could effectively promote osteoblast proliferation and osteogenic differentiation by the activation of the *TAZ*/*RUNX2* signaling pathway [37]. Additionally, our previous studies also found a facile means to engineer oriented surface topology of poly(ε-caprolactone) (PCL) by combing uniaxially stretching the PCL substrate with homoepitaxial crystallization [58]. Surface epiphytic crystallization is a chemical method that can make the attached biomass produce an unusual topological structure on the surface of an object and is applied to adjust the surface morphology of semi-crystalline polymers.

In our study, we successfully fabricated a layer of nanoridge on the surface of 3D-printed scaffolds via surface epiphytic crystallization [59]. After surface modification, the surface roughness of the scaffold is significantly increased. The hemolysis test and animals’ skin stimulation results of the BPS and TPS were negative. Additionally, infiltration media of BPS and TPS exhibit no cytotoxicity to USCs. The thickness of the patterned nanosheets constructed in this study is about tens of nanometers, so it has no significant effect on the overall pore size and porosity of the scaffold. The results of FTIR and XPS demonstrated that the surface epidermal crystals did not change the elemental composition of the original matrix material. All these results confirmed that surface epiphytic crystallization was a safe and effective patterning method of 3D printed PCL scaffolds. Unlike traditional surface patterning technologies, such as photolithography, electron beam etching, laser particle beam etching, and nanoimprint technology, which could only construct two-dimensional surface patterning, surface epiphytic crystallization can be used for the interior and surface of three-dimensional scaffolds patterning construction. Simultaneously, compared with the disadvantages of traditional patterning technology such as high cost and complicated process, surface epiphytic crystallization has the advantages of low cost and simple process, and is easier to be used for application in further research. Additionally, after the implantation of scaffolds into the body, various proteins will attach to the surface of scaffolds from the blood and other body fluids, which is prior to the attachment of cells. Evaluating the protein adsorption behavior of the scaffolds plays a vital role in assessing the bioactivity of the surface of scaffolds and the way how the scaffolds regulate cell behaviors. Our study demonstrated that constructing a layer of patterned nanoridges on the surface of 3D-printed PCL scaffolds could significantly improve the protein adsorption capacity of the scaffold. In vitro mineralization studies have found that patterned nanoridges can promote the deposition of hydroxyapatite on the surface of the scaffold, which has also been confirmed by in vivo results where the presence of patterned nanoridges improved the bone repair ability of the scaffolds.

Cell growth and functions in bone regeneration are tightly associated with the cellular interactions at the cell−extracellular matrix (ECM) biointerface [60]. Increasing evidence indicates that the cellular response to environmental signaling goes far beyond the biochemical cues [61]. Physical cues, especially the surface topographic features, have gradually been recognized as key factors that mediate many cell behaviors, including cell attachment, proliferation, migration, and differentiation, by activating related signal pathways [62]. As an important property of surface topography, the surface roughness of scaffolds can be presented in random patterning or regular patterning. The surface roughness of scaffolds can directly regulate the migration, extension, arrangement, proliferation, and differentiation of seed cells on the surface of scaffolds [63]. Our study confirmed that the layer of nanoridge on the surface of 3D-printed scaffolds could effectively enhance the USCs proliferation and osteogenic gene expression of USCs via enhancing the surface roughness of scaffolds. Simultaneously, the in vivo results also confirmed that TPS with a rough surface has better bone repair ability in vivo than BPS with a smooth surface. The actin cytoskeleton is a crucial determinant of cell shape, which can be more precisely explained as the assembly and disassembly of actin filaments [64]. Notably, actin cytoskeleton-mediated cell shape changes have been shown to be vital for the regulation of MSC lineage commitment [64]. Our study found that compared to the appearance of round USCs, long rod-shaped USCs have better osteogenic differentiation ability, which is consistent with previous studies [24,65]. At present, several studies confirmed that the regulation of osteogenesis mediated by the cytoskeleton or cell shape is related to the activation of RhoA/ROCK and YAP/TAZ mechanosensitive signaling pathways [66,67].

PCL is a kind of polymer material with good biocompatibility, which is currently widely applied as a cell carrier in the field of tissue regeneration. In addition, PCL is also a common raw material for constructing 3D printed scaffolds. However, most 3D printed PCL-based polymer scaffolds possess flat surface topography, which cannot effectively regulate cell behavior. Our research effectively improved the cell regulation ability of the scaffolds by modulating surface topographic features. The in vivo results confirmed that the group of TPS loaded with USCs has better bone repair ability than that of other groups. Notably, the in vivo biosafety evaluation results confirmed that TPS loaded with USCs was also a safe way of bone regeneration in vivo.

Although TPS loaded with USCs was the most effective treatment in this study, it was encouraging as a tissue engineering scaffold that TPS without stem cells had a similar effect to BPS loaded with stem cells. Stem cells, scaffolds, and growth factors are three basic elements in tissue engineering or regenerative medicine [68]. The repair ability of implanted stem cells can be influenced by the properties of scaffolds. The ideal scaffolds for tissue engineering should possess a good capacity to support various biological behaviors of stem cells. In our opinion, the repair ability of USCs has been partly inhibited because the BPS could not provide a suitable micro-environment for USCs. In addition, applying scaffolds without using stem cells is another common therapeutic strategy for tissue regeneration [69]. The chemokines secreted from injury tissues can recruit endogenous stem cells into the injury site to take part in the process of various tissue regeneration [70]. In our study, the bioactive interface of a scaffold provides a suitable micro-environment for endogenous stem cells, resulting in the good repair ability of TPS without USCs.

There are several limitations to our study. Firstly, USCs can be considered to be an alternative to traditional stem cells for bone regeneration, but the osteogenic difference in vivo and in vitro between USCs and other kinds of MSCs in vivo and in vitro needs to be further explored. Secondly, the low immunogenicity of stem cells has been demonstrated by many previous studies, but the fate of stem cells implanted in the body still remains controversial. Cell labeling and cell tracking of stem cells are needed for further research in future. Thirdly, actin filament remodeling appears to be a vital determinant in the differentiation of MSCs. However, further studies are needed to investigate how actin contributes to osteogenesis and what is the role of nuclear actin in MSCs differentiation.

## 5. Conclusions

In this study, we have successfully fabricated a nanoridge layer on the surface of a 3D-printed PCL scaffold via that surface epiphytic crystallization. Modulation of surface topographic features effectively enhance surface roughness, protein adsorption capacity, and mineralization in vitro of scaffolds. Simultaneously, we have successfully isolated an ideal kind of stem cell, USCs, from adult urine by a non-invasive process. The existence of a nanoridge layer could promote the proliferation and osteogenic-related gene expression of USCs. Additionally, TPS loaded with USCs is a safe and effective method for bone defect repair in vivo, which provides the theoretical basis for the clinical application of 3D-printed bone substitutes in bone regeneration.

## Figures and Tables

**Figure 1 pharmaceutics-14-01437-f001:**
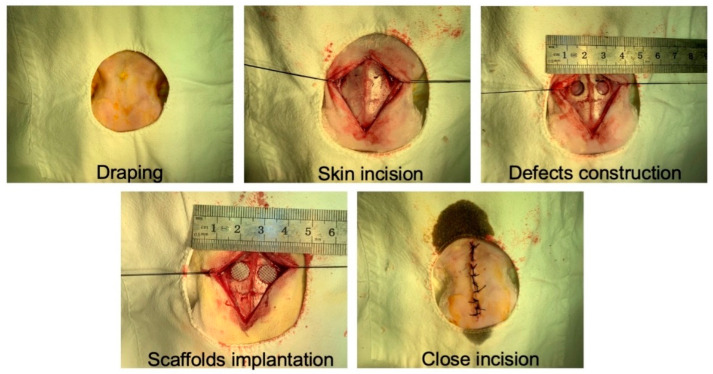
Images of rabbit cranial bone defect preparation.

**Figure 2 pharmaceutics-14-01437-f002:**
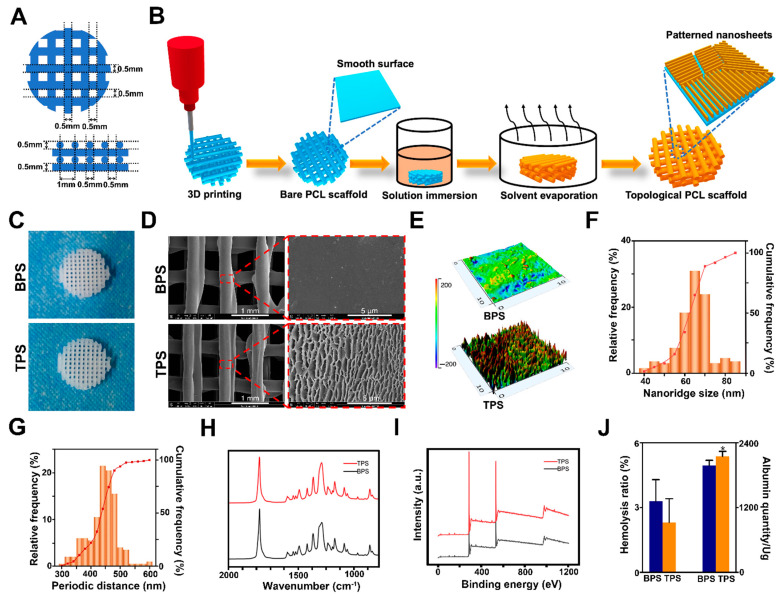
(**A**) Two-dimensional schematic diagram of 3D-printed PCL scaffolds. (**B**) Procedure of surface topological modulation of 3D-printed PCL scaffolds. (**C**) General view of TPS and BPS. (**D**) SEM scanning of TPS and BPS. (**E**) AFM view of TPS and BPS. (**F**) The distribution of nanoridge size of TPS. (**G**) The distribution of periodic distance of TPS. (**H**) The FTIR results of TPS and BPS. (**I**) The XPS results of TPS and BPS. (**J**) The hemolysis ratio and albumin adsorption quantity of TPS and BPS. * *p* < 0.05.

**Figure 3 pharmaceutics-14-01437-f003:**
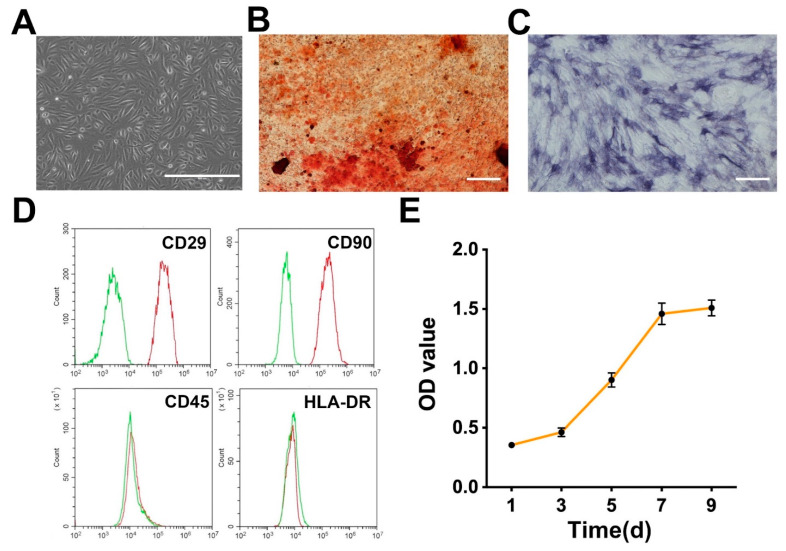
(**A**) The cellular morphology of USCs. Scale bar: 400 μm. (**B**) The Alizarin Red staining of USCs. Scale bar: 200 μm. (**C**) The alkaline phosphatase staining of USCs. Scale bar: 200 μm. (**D**) The flow cytometric analysis of USCs. The red and green lines represent the experiment and control groups, respectively. (**E**) The growth curve of USCs.

**Figure 4 pharmaceutics-14-01437-f004:**
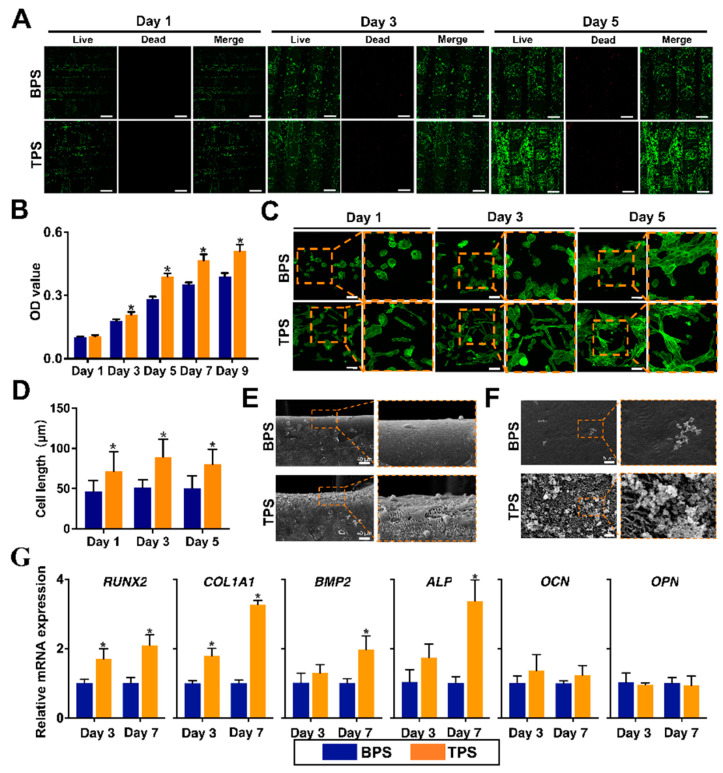
(**A**) Live/Dead staining on day 1, day 3, and day 5. Scale bar: 500 μm. (**B**) The CCK-8 results of USCs on the surface of BPS and TPS on day 1, day 3, day 5, day 7, and day 9. * *p* < 0.05. (**C**) The Phalloidin staining of USCs on BPS and TPS on day 1, day 3, and day 5. Scale bar: 50 μm. (**D**) The cell lengths of USCs on the surface of BPS and TPS. * *p* < 0.05. (**E**) The SEM images of BPS and TPS loaded with USCs. Scale bar: 40 μm. (**F**) The SEM images of BPS and TPS immersed in SBF. Scale bar: 5 μm. (**G**) The *RUNX2*, *COL1A1*, *BMP2*, *ALP*, *OCN*, and *OPN* mRNA expression of USCs on the surface of BPS and TPS on day 3 and day 7. * *p* < 0.05.

**Figure 5 pharmaceutics-14-01437-f005:**
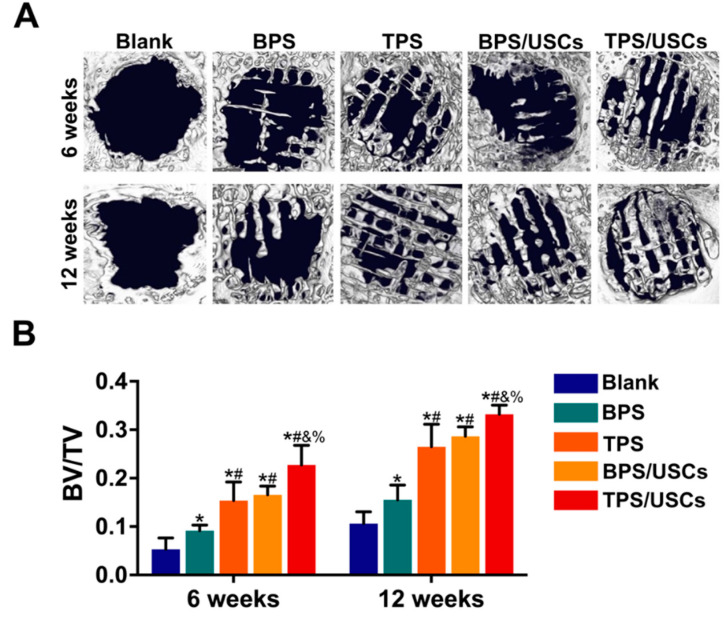
(**A**) The Micro-CT images of each group at six and twelve weeks after surgery. (**B**) The BV/TV of each group at six and twelve weeks after surgery. Bone Volume/Total Volume, BV/TV. * *p* < 0.05, compared with blank group. # *p* < 0.05, compared with BPS group. & *p* < 0.05, compared with TPS group. % *p* < 0.05, compared with a group of BPS loaded with USCs.

**Figure 6 pharmaceutics-14-01437-f006:**
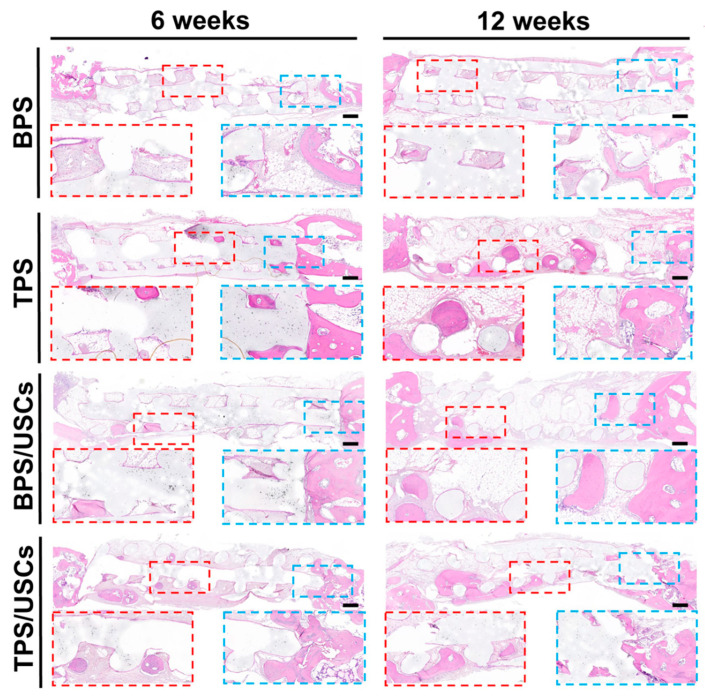
The HE staining results of the central area of the bone defects at six and twelve weeks after implantation. Scale bar: 500 μm. The red and blue squares represent the magnification view of part region in the HE staining.

**Figure 7 pharmaceutics-14-01437-f007:**
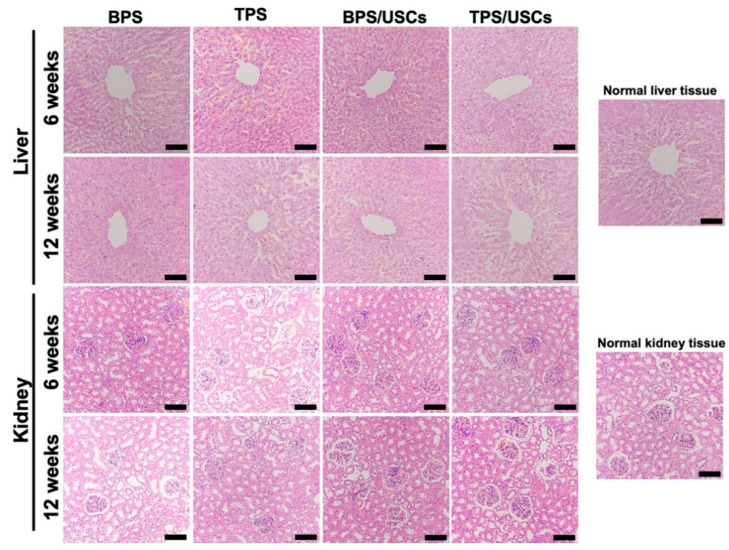
The HE staining results of liver and kidney tissue at six and twelve weeks after operation. Scale bar: 100 μm.

**Figure 8 pharmaceutics-14-01437-f008:**
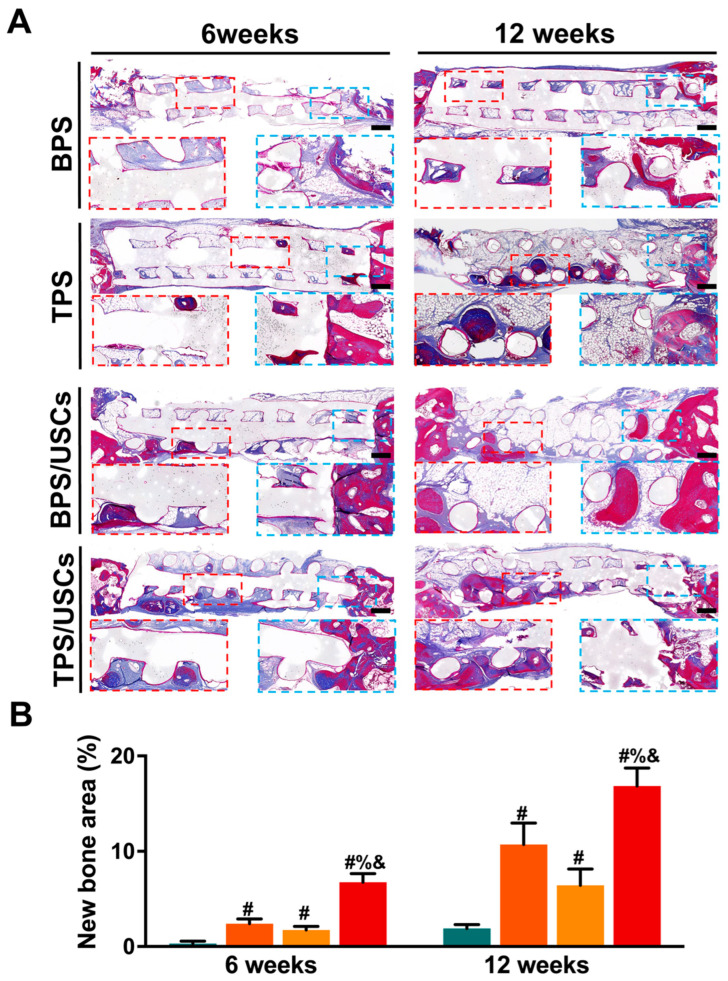
(**A**) The Masson staining results of the central area of the bone defects at six and twelve weeks after implantation. Scale bar: 500 μm. The red and blue squares represent the magnification view of part region in the Masson staining. (**B**) The new bone area of each group at six and twelve weeks after surgery. # *p* < 0.05, compared with BPS group. & *p* < 0.05, compared with TPS group. % *p* < 0.05, compared with a group of BPS loaded with USCs.

**Table 1 pharmaceutics-14-01437-t001:** Primers for real-time polymerase chain reaction.

Target Gene	Forward Primer Sequence (5′-3′)	Reverse Primer Sequence (5′-3′)	Length of Amplicon
*GAPDH*	ACAACTTTGGTATCGTGGAAGG	GCCATCACGCCACAGTTTC	91
*BMP2*	ACCCGCTGTCTTCTAGCGT	TTTCAGGCCGAACATGCTGAG	180
*RUNX2*	CCAACCCACGAATGCACTATC	TAGTGAGTGGTGGCGGACATAC	91
*ALP*	ACCACCACGAGAGTGAACCA	CGTTGTCTGAGTACCAGTCCC	79
*OCN*	CCCCCTCTAGCCTAGGACC	ACCAGGTAATGCCAGTTTGC	169
*COL1A1*	GCCCAGAAGAACTGGTACATCAG	CGCCATACTCGAACTGGAATC	97
*OPN*	GCCGAGGTGATAGTGTGGTT	TGAGGTGATGTCCTCGTCTG	101

## Data Availability

Not applicable.

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
