# Peer review of "Nanotopographical 3D-Printed Poly(ε-caprolactone) Scaffolds Enhance Proliferation and Osteogenic Differentiation of Urine-Derived Stem Cells for Bone Regeneration"

_pharmaceutics, 2022, doi:10.3390/pharmaceutics14071437_

Round 1

Reviewer 1 Report

Dear Authors,

 Thank you for submitting the manuscript entitled "Nanotopographical 3D-Printed Poly(Ɛ-caprolactone) Scaffolds enhance proliferation and osteogenic differentiation of Urine-derived Stem Cells for bone regeneration."

In this paper, authors describe the development of a nanotopographical poly(Ɛ-caprolactone) scaffolds via 3D printing technology. Authors reported that these scaffolds loaded with human stem cells isolated from urine of healthy adult male donors is a safe and effective therapeutic strategy for bone regeneration.

The paper has some strengths, but there are some critical points that need clarification and improvement especially concerning gene expression analyses.

 Introduction

The first part of the Introduction section should be improved, especially regarding the rationale behind the use of tissue engineered bone substitutes for the treatment of bone defects.

Authors should better describe the current available options to treat bone defects taking into account that the number of tissue engineered products reaching the market is limited.

To this aim, the following references may help the authors to better focus the field:

Govoni M, Vivarelli L, Mazzotta A, Stagni C, Maso A, Dallari D. Commercial Bone Grafts Claimed as an Alternative to Autografts: Current Trends for Clinical Applications in Orthopaedics. Materials (Basel). 2021 Jun 14;14(12):3290. doi: 10.3390/ma14123290.

Grassi FR, Grassi R, Vivarelli L, Dallari D, Govoni M, Nardi GM, Kalemaj Z, Ballini A. Design Techniques to Optimize the Scaffold Performance: Freeze-dried Bone Custom-made Allografts for Maxillary Alveolar Horizontal Ridge Augmentation. Materials (Basel). 2020 Mar 19;13(6):1393. doi: 10.3390/ma13061393.

The overlapping fields of tissue engineering and regenerative medicine date to the late 1980s and early 1990s, suggesting that the field has been studied from about 30 years. Nevertheless, a very limited number of TE products (intended as cellular constructs cultivated in static/dynamic conditions) have reached the market.

In 2006, Dr Nerem (Tissue Eng. 2006 May;12(5):1143-50. doi: 10.1089/ten.2006.12.1143) reported that tissue engineering was overpromised and underdelivered.

Please provide an update concerning the hype and real expectations of BTE protocols nowadays, especially considering that a take-home message from this paper is that nanotopographical 3D-printed scaffolds loaded with USCs is a safe and effective therapeutic strategy for bone regeneration.

Moreover, since the other topic of this work is the 3D printing of nanotopographical scaffolds, authors should better detail the final part of the Introduction section adding further information and appropriate references concerning the issues related to development of specific topological morphology on the surface of complex 3D-objects. In this respect, mastering this technology requires the setting of specific parameters for patterning biocompatible materials to manufacture 3D constructs with mechanical and biological properties suitable for the deposition of living cells.

 The following references may help the authors to better focus the field:

Lovecchio J. et al. Fiber Thickness and Porosity Control in a Biopolymer Scaffold 3D Printed through a Converted Commercial FDM Device. Materials (Basel). 2022 Mar 24;15(7):2394. doi: 10.3390/ma15072394.

Saini, G.; Segaran, N.; Mayer, J.L.; Saini, A.; Albadawi, H.; Oklu, R. Applications of 3D Bioprinting in Tissue Engineering and Regenerative Medicine. J. Clin. Med. 2021, 10, 4966.

Chang, C.C.; Boland, E.D.; Williams, S.K.; Hoying, J.B. Direct-write bioprinting three-dimensional biohybrid systems for future regenerative therapies. J. Biomed. Mater. Res. B Appl. Biomater. 2011, 98, 160–170.

Materials and Methods

Throughout the manuscript please replace ml with mL.

Moreover, refer to the following reference for the correct nomenclature to use for human genes:

Bruford, E. A., Braschi, B., Denny, P., Jones, T., Seal, R. L., & Tweedie, S. (2020). Guidelines for human gene nomenclature. Nature genetics, 52(8), 754–758. https://doi.org/10.1038/s41588-020-0669-3.

Lines 120-121: Please provide specific information regarding where the keratinocyte serum-free medium and embryo fibroblast medium were purchased (company, city, country).

Line 137: Please replace 2x105/mL with 2x105 mL

Line 151: Please note that -2DDCT is an incorrect method to calculate the relative changes in gene expression. The correct method is 2-DDCq.

This reviewer hopes that authors have used the correct method for their gene expression analyses.

Line 146: Please replace qRT-PCR with RT-qPCR.

Moreover, this reviewer strongly recommends following the MIQE guidelines (The MIQE guidelines: minimum information for publication of quantitative real-time PCR experiments. Clin Chem. 2009 Apr;55(4):611-22. doi: 10.1373/clinchem.2008.112797), not only to use the correct nomenclature in the text, but also to enable other investigators to reproduce results. In this respect, please replace CT values with Cq values, and housekeeping gene with reference gene.

The use of reference genes as internal controls is the most common method for normalizing cellular mRNA data. However, the reference gene should be stably expressed between treatment groups.

Hence, normalization against a single reference gene (e.g., GAPDH), as reported by the authors, is not acceptable unless the investigators present clear evidence that confirms its invariant expression under the experimental conditions described.

To this regard, has the M-value (the measure of expression stability) of your reference gene been calculated?

Have the authors purchased certified primers? In this case, please add information regarding the Company. On the other hand, if primers were custom designed, please add the length of amplicons in Table 1.

Author Response

Dear editor and reviewer:

Thank you very much for your comments on our manuscript! Your comments were highly insightful and enabled us to greatly improve the quality of our manuscript. Our point-by-point responses to each of the comments are listed below.

Comment 1:  The first part of the Introduction section should be improved, especially regarding the rationale behind the use of tissue engineered bone substitutes for the treatment of bone defects.Authors should better describe the current available options to treat bone defects taking into account that the number of tissue engineered products reaching the market is limited.To this aim, the following references may help the authors to better focus the field: Govoni M, Vivarelli L, Mazzotta A, Stagni C, Maso A, Dallari D. Commercial Bone Grafts Claimed as an Alternative to Autografts: Current Trends for Clinical Applications in Orthopaedics. Materials (Basel). 2021 Jun 14;14(12):3290. doi: 10.3390/ma14123290. Grassi FR, Grassi R, Vivarelli L, Dallari D, Govoni M, Nardi GM, Kalemaj Z, Ballini A. Design Techniques to Optimize the Scaffold Performance: Freeze-dried Bone Custom-made Allografts for Maxillary Alveolar Horizontal Ridge Augmentation. Materials (Basel). 2020 Mar 19;13(6):1393. doi: 10.3390/ma13061393.

Response 1: Thanks for your kind comments. We have added related content in Line 34-42. Please see text highlighted in red in the manuscript. In addition, we also cited these two important references in our manuscript.

Comment 2: The overlapping fields of tissue engineering and regenerative medicine date to the late 1980s and early 1990s, suggesting that the field has been studied from about 30 years. Nevertheless, a very limited number of TE products (intended as cellular constructs cultivated in static/dynamic conditions) have reached the market.In 2006, Dr Nerem (Tissue Eng. 2006 May;12(5):1143-50. doi: 10.1089/ten.2006.12.1143) reported that tissue engineering was overpromised and underdelivered. Please provide an update concerning the hype and real expectations of BTE protocols nowadays, especially considering that a take-home message from this paper is that nanotopographical 3D-printed scaffolds loaded with USCs is a safe and effective therapeutic strategy for bone regeneration.

Response 2: Thanks for your kind comments. We have added related content in Line 42-53. Please see text highlighted in red in the manuscript.

Comment 3: Moreover, since the other topic of this work is the 3D printing of nanotopographical scaffolds, authors should better detail the final part of the Introduction section adding further information and appropriate references concerning the issues related to development of specific topological morphology on the surface of complex 3D-objects. In this respect, mastering this technology requires the setting of specific parameters for patterning biocompatible materials to manufacture 3D constructs with mechanical and biological properties suitable for the deposition of living cells.The following references may help the authors to better focus the field: Lovecchio J. et al. Fiber Thickness and Porosity Control in a Biopolymer Scaffold 3D Printed through a Converted Commercial FDM Device. Materials (Basel). 2022 Mar 24;15(7):2394. doi: 10.3390/ma15072394. Saini, G.; Segaran, N.; Mayer, J.L.; Saini, A.; Albadawi, H.; Oklu, R. Applications of 3D Bioprinting in Tissue Engineering and Regenerative Medicine. J. Clin. Med. 2021, 10, 4966. Chang, C.C.; Boland, E.D.; Williams, S.K.; Hoying, J.B. Direct-write bioprinting three-dimensional biohybrid systems for future regenerative therapies. J. Biomed. Mater. Res. B Appl. Biomater. 2011, 98, 160–170.

Response 3: That is a great suggestion. We have added related content in Line 87-96. Please see text highlighted in red in the manuscript. In addition, we also cited these three important references in our manuscript. Thanks for your kind comments.

Comment 4: Throughout the manuscript please replace ml with mL.

Response 4: Thanks for your kind comments. We have replaced ml with mL and checked the manuscript. Please see text highlighted in red in the manuscript.

Comment 5: Moreover, refer to the following reference for the correct nomenclature to use for human genes:Bruford, E. A., Braschi, B., Denny, P., Jones, T., Seal, R. L., & Tweedie, S. (2020). Guidelines for human gene nomenclature. Nature genetics, 52(8), 754–758. https://doi.org/10.1038/s41588-020-0669-3.

Response 5: Thanks for your kind comments. We have revised the manuscript according to suggestions.  Please see text highlighted in red in the manuscript.

Comment 6: Lines 120-121: Please provide specific information regarding where the keratinocyte serum-free medium and embryo fibroblast medium were purchased (company, city, country).

Response 6: We have revised the manuscript according to suggestions.  Please see text highlighted in red in the manuscript (Line 136-137). Thanks for your kind comments.

Comment 7: Line 137: Please replace 2x105/mL with 2x105 mL

Response 7: We have revised the manuscript according to suggestions.  Please see text highlighted in red in the manuscript (Line 152). Thanks for your kind comments.

Comment 8: Line 151: Please note that -2DDCT is an incorrect method to calculate the relative changes in gene expression. The correct method is 2-DDCq. This reviewer hopes that authors have used the correct method for their gene expression analyses.

Response 8: We have revised the manuscript according to suggestions.  Please see text highlighted in red in the manuscript (Line 167). Thanks for your kind comments.

Comment 9: Line 146: Please replace qRT-PCR with RT-qPCR. Moreover, this reviewer strongly recommends following the MIQE guidelines (The MIQE guidelines: minimum information for publication of quantitative real-time PCR experiments. Clin Chem. 2009 Apr;55(4):611-22. doi: 10.1373/clinchem.2008.112797), not only to use the correct nomenclature in the text, but also to enable other investigators to reproduce results. In this respect, please replace CT values with Cq values, and housekeeping gene with reference gene.

Response 9: We have revised the manuscript according to suggestions.  We also revised Figure 4. Please see text highlighted in red in the manuscript (Line 159-160). Thanks for your kind comments.

Comment 10: The use of reference genes as internal controls is the most common method for normalizing cellular mRNA data. However, the reference gene should be stably expressed between treatment groups.Hence, normalization against a single reference gene (e.g., GAPDH), as reported by the authors, is not acceptable unless the investigators present clear evidence that confirms its invariant expression under the experimental conditions described. To this regard, has the M-value (the measure of expression stability) of your reference gene been calculated?

Response 10: Thanks for your kind comments. We have already revised the part of gene expression according to your kind suggestions. In addition, GAPDH is the most common reference gene used in tissue engineering. Many previous studies have demonstrated that GAPDH is stable expression in osteogenesis analysis, which is consistent with our previous research in our lab.

Comment 11: Have the authors purchased certified primers? In this case, please add information regarding the Company. On the other hand, if primers were custom designed, please add the length of amplicons in Table 1.

Response 11: Thanks for your kind comments. We have already revised the manuscript and added the length of amplicons in Table 1.

We really hope that the revisions in the manuscript and our accompanying responses will be sufficient to make our manuscript suitable for publication in Pharmaceutics. If you have any questions, please do not hesitate to contact me.

Best wishes

Xin Duan and Jia-Zhuang Xu
Orthopedic Research Institute, Department of Orthopedics, West China Hospital, Sichuan University, No. 37 Guoxue Lane, Chengdu 610041, Sichuan, China

College of Polymer Science and Engineering and State Key Laboratory of Polymer Materials Engineering, Sichuan University, Chengdu 610065, Sichuan, China

Reviewer 2 Report

In this study, authors proposed nanotopographical 3D-Printed polycaprolactone scaffolds to improve proliferation and osteogenic differentiation of urine derived stem cells. Their scaffolds with stem cells enhanced the new bone formation in rabbit cranial defect model. This article was well organized and sentences were cleared. If some contents are revised, it will be possible to publish in the Pharmaceutics.

1. In the case of the bone formation ratio in Figure 5 and the new bone area graph in Figure 8B, the TPS with a nano-surface showed very good performance, apart from the superiority of TPS/USCs. Although TPS/USCs was the most effective conditions in this study, it was encouraging as tissue engineering scaffolds that TPS without stem cells had a similar effect to study (BPS/USCs) using stem cells. Therefore, if additional descriptions and discussions about the comparison between TPS and BPS/USCs are added in section 3.5 and discussion (section 4), the quality of the manuscript will be improved.

2. In Figure 3E, it is necessary to display the raw data (OD values) from day 1 to day 9 to obtain the growth curve data.

3. In the live/dead assay of Figure 4A, it is required to add a survival rate graph to confirm that which percentage of the cells were lived over time.

4. In the graph of Figure 8B, the index for each condition (BPS, TPS, BPS/USCs, TPS/USCs) is omitted. It is required to add the index.

5. In Figure 4, Figure 6, Figure 7 and Figure 8 images, there was no scale bar or there was no scale information in the figure caption. It is required to add the information.

Author Response

Dear editor and reviewer:

Thank you very much for your comments on our manuscript! Your comments were highly insightful and enabled us to greatly improve the quality of our manuscript. Our point-by-point responses to each of the comments are listed below.

Comment 1:  In the case of the bone formation ratio in Figure 5 and the new bone area graph in Figure 8B, the TPS with a nano-surface showed very good performance, apart from the superiority of TPS/USCs. Although TPS/USCs was the most effective conditions in this study, it was encouraging as tissue engineering scaffolds that TPS without stem cells had a similar effect to study (BPS/USCs) using stem cells. Therefore, if additional descriptions and discussions about the comparison between TPS and BPS/USCs are added in section 3.5 and discussion (section 4), the quality of the manuscript will be improved.

Response 1: That is a great suggestion. We have added related content in Line 372-374 and Line 484-497. Please see text highlighted in red in the manuscript. Thanks for your kind comments.

Comment 2: In Figure 3E, it is necessary to display the raw data (OD values) from day 1 to day 9 to obtain the growth curve data.

Response 2: We have revised the Figure 3E according to your kind suggestions. Thanks for your kind comments.

Comment 3: In the live/dead assay of Figure 4A, it is required to add a survival rate graph to confirm that which percentage of the cells were lived over time.

Response 3: That is a great suggestion. We have revised the Figure according to your kind suggestions. Thanks for your kind comments. Please see the Figure S1D. Thanks for your kind comments.

Comment 4: In the graph of Figure 8B, the index for each condition (BPS, TPS, BPS/USCs, TPS/USCs) is omitted. It is required to add the index.

Response 4: Thanks for your kind comments. We have revised Figure 8B according to your suggestions.

Comment 5: In Figure 4, Figure 6, Figure 7 and Figure 8 images, there was no scale bar or there was no scale information in the figure caption. It is required to add the information.

Response 5: Thanks for your kind comments. We have revised the Figure 4, Figure 6, Figure 7 and Figure 8. In addition, we have added the information of scale bar in figure notes. Please see text highlighted in red in the manuscript.

We really hope that the revisions in the manuscript and our accompanying responses will be sufficient to make our manuscript suitable for publication in Pharmaceutics. If you have any questions, please do not hesitate to contact me.

Best wishes

Xin Duan and Jia-Zhuang Xu
Orthopedic Research Institute, Department of Orthopedics, West China Hospital, Sichuan University, No. 37 Guoxue Lane, Chengdu 610041, Sichuan, China

College of Polymer Science and Engineering and State Key Laboratory of Polymer Materials Engineering, Sichuan University, Chengdu 610065, Sichuan, China

Reviewer 3 Report

This is a well written paper about application of 3D printing and stem cells for tissue repair. I suggest Editor to accept the paper. These are minor suggestions for authors:

1. Please provide more details about 3D printing process. Based on the details provided in the paper, it seems that authors used Melt Extrusion Deposition (MED™) 3D printing.

2. There are some typographic errors: nanorige or nanoridge; Figure designations should be placed before the end of sentence. Measuring units should be separate with a space from the numbers. There is an extra space in line 319.

Author Response

Dear editor and reviewer:

Thank you very much for your comments on our manuscript! Your comments were highly insightful and enabled us to greatly improve the quality of our manuscript. Our point-by-point responses to each of the comments are listed below.

Comment 1:  Please provide more details about 3D printing process. Based on the details provided in the paper, it seems that authors used Melt Extrusion Deposition (MED™) 3D printing.

Response 1: We have added related content in Line 101-112. Please see text highlighted in red in the manuscript. Thanks for your kind comments.

Comment 2: There are some typographic errors: nanorige or nanoridge;

Response 2: We have revised the manuscript according to your kind suggestions. Please see text highlighted in red in the manuscript. Thanks for your kind comments.

Comment 3: Figure designations should be placed before the end of sentence.

Response 3: We have revised the manuscript according to your kind suggestions. Please see text highlighted in red in the manuscript. Thanks for your kind comments.

Comment 4: Measuring units should be separate with a space from the numbers.

Response 4: Thanks for your kind comments. We have revised the manuscript according to your suggestions.

Comment 5: There is an extra space in line 319.

Response 5: Thanks for your kind comments. We have revised the manuscript according to your suggestions.

We really hope that the revisions in the manuscript and our accompanying responses will be sufficient to make our manuscript suitable for publication in Pharmaceutics. If you have any questions, please do not hesitate to contact me.

Best wishes

Xin Duan and Jia-Zhuang Xu
Orthopedic Research Institute, Department of Orthopedics, West China Hospital, Sichuan University, No. 37 Guoxue Lane, Chengdu 610041, Sichuan, China

College of Polymer Science and Engineering and State Key Laboratory of Polymer Materials Engineering, Sichuan University, Chengdu 610065, Sichuan, China

Reviewer 4 Report

Thanks to the authors for sharing their research findings on surface modified 3d printed PCL scaffolds and its effects on USCs. The manuscript is well presented. However, please provide explanation for following questions to improve manuscript's quality.

Please include citations for MSC markers for flow cytometry study in the methods/results.

Provide IRB approval details for urine sample collection from individuals. Also mention selection criteria.

Provide ethical approval for animal study

Provide details regarding scaffold fabrication for the test groups containing USCs. After adding 100 ul of cell suspension, were the scaffolds immediately placed in the animals or were they cultured further.

Correct inconsistency in the bibliography-

Line 502 reference  has doi while others don't.

page numbers missing for a few reference 

Round 2

Reviewer 1 Report

The authors addressed all comments and improved the quality of the manuscript.

However, during the paste and copy of my previous comments in the SuSy platform, the symbol of delta was lost.

Please replace 2-DDCq with 2-DDCq at line 168.